# Reflections of Moral Suffering, Resilience, and Wisdom of Pediatric Oncology Social Workers during the COVID-19 Pandemic

Barbara Jones [1], Nancy Cincotta [2], Wendy Pelletier [3], Abigail Fry [4] and Lori Wiener [4,*]

1   Steve Hicks School of Social Work, University of Texas at Austin, Austin, TX 78712, USA
2   School of Social Work, Columbia University, New York, NY 10027, USA
3   Alberta Children's Hospital, Calgary, AB T3B 6A8, Canada
4   National Cancer Institute, Center for Cancer Research, National Institutes of Health, Bethesda, MD 20814, USA
*   Correspondence: wienerl@mail.nih.gov

**Abstract:** Background: The COVID-19 pandemic has significantly altered the lives of pediatric oncology social workers. Challenges include difficulty building rapport with the use of telephone/computers, lack of clarity around who is designated as "essential", structural challenges, isolation, and witnessing distress. This study aimed to describe the ways that the pandemic has personally impacted pediatric oncology social workers. Methods: Participants were recruited through the Association of Pediatric Oncology Social Workers (APOSW) listserv. In total, 101 participants from 31 states and the District of Columbia completed an online survey containing quantitative and open-ended questions. Qualitative data analysis included thematic analysis of participants' optional survey responses to three open-ended questions. Results: Fifty-seven of the participants provided responses that revealed 3 first level codes and 11 second level codes. First level codes were developed a priori from the questions: Experiences that stay with you, Wisdom gained and Impact on your work. Pandemic-related challenges caused moral suffering and professional challenges for participants but also created opportunities to find meaning in their work. Conclusion: Data illuminated moral suffering, unrecognized resilience, new ways of maintaining self-and family care, and creative approaches to care of children with cancer and their families at diagnosis, during treatments and at the end of life.

**Keywords:** COVID-19; social work practice; pediatrics; psychosocial; qualitative methods; moral suffering; resilience

## 1. Introduction

In December of 2019, the world became aware of a novel coronavirus and in March of 2020 the World Health Organization declared the novel coronavirus (SARS-CoV-2) to be a pandemic. By February of 2021, over 106 million cases had been diagnosed worldwide with more than 2 million deaths. One year later, that number more than tripled with over 400 million cases and almost 6 million deaths. Despite the availability of vaccines against COVID-19 since December of 2022 and currently over 4 billion people now vaccinated worldwide, cases and deaths have continued to occur [1] (https://coronavorus.jhu.edu, accessed on 14 March 2020). Although vaccines against COVID-19 have provided hope for a potential path forward to de-escalate the crisis and allow some return to normalcy, the impact of the pandemic on the entire world is undeniable and continuing.

In the early months of the pandemic, prior to vaccine availability and delivery, institutional crises and lockdowns brought a devastating new reality to healthcare settings. For healthcare providers, including pediatric oncology social workers, their personal lives as well as professional practices were significantly challenged [2].

Professionally, dramatic changes in roles, responsibilities, working remotely, the shift to electronic service provision, and functioning often with a perception of decreased support

from their institutions, has contributed to moral distress [3,4]. Social workers and other healthcare professionals working with pediatric patients at end of life, and their families, may be even more impacted by the changes and stresses associated with COVID-19 [5]. Considering professional social work values and ethics around patient and family centered self-determination, social workers have reported challenges to maintaining confidentiality, offering privacy, providing patient access to virtual technology, and holding professional boundaries [4].

Health care providers have reported a personal psychological toll resulting in burnout, emotional trauma, and an increase in mental health disorders [6]. Fears related to transmitting the virus to loved ones, patients, and colleagues, as well as guilt, fear, exhaustion, and anxiety around loss of income have all been contributing factors [7]. In some instances where social workers and other healthcare professionals were tasked with enforcing infection control protocols, they became the target of verbal aggression from families. This contributed to a sense of fatigue, burnout, fear, and anxiety [8].

It has been suggested that social workers have had, and indeed still have a "herculean" role to play in reframing social work practices that embrace new models of service delivery including virtual platforms that may persist post-pandemic [9]. Advocating for marginalized and diverse populations who may not have access to technology and may suffer because of not physically attending clinics where they have increased support of staff has been reported [2]. Hybrid models of telehealth/in-person care, particularly when working with seriously ill children were contemplated and implemented [10].

A cross-sectional online survey was designed to formally study the impact of the COVID-19 pandemic on the professional and personal lives of pediatric oncology social workers [2]. The survey included open-ended questions designed to explore in more detail the unique experiences, wisdom gained and impact on work reported by pediatric oncology social workers.

## 2. Materials and Methods

### 2.1. Design and Sample

This study was a national, cross-sectional survey-based study of pediatric oncology social workers in the United States. The anonymous survey was posted with permission on the Association of Pediatric Oncology Social Workers (APOSW) listserv after review and approval by the APOSW research committee. Participants were asked to complete the survey if they were currently working as a pediatric oncology social worker and were invited to share the survey with their pediatric oncology social work colleagues. Between 5 October and 20 November 2020, the listserv posted one announcement with two follow-up reminders, spaced 14 days after the initial announcement.

### 2.2. Measures

The study instrument was adapted from a survey of pediatric palliative care professionals that was designed by a collaborative, interdisciplinary study team according to the Tailored Method of Survey Design [5,11]. We tailored the survey questions to address issues specific to pediatric oncology social workers and COVID-19. The final survey instrument was comprised of 57 multiple choice questions, several with an option to describe their choice, and three open-ended questions.

The quantitative survey questions, reported elsewhere [2], were focused on social work clinical practice prior to and during COVID-19, team structure, and patient care including through end-of-life and bereavement. As part of the larger study, three open-ended questions which had been previously and successfully used with pediatric palliative care providers [12] were approved for use in this study and are reported here. *(1) Can you tell us about an experience you have had related to COVID-19 that you feel will stay with you, always? (2) What wisdom have you gained from living and working through the pandemic that would be helpful to future social workers dealing with similar situations? Please elaborate. (3) Please*

*take this opportunity to share any other ways that your work has been impacted by COVID-19 that have not been captured in this survey.*

### 2.3. Data Collection and Analysis

The Office of Human Subjects Research Protections at the National Institutes of Health determined that the survey format and content qualified as exempt from full Institutional Review Board review and consent was waived. An encrypted SurveyMonkey© (San Mateo, CA, USA) questionnaire was utilized for online data collection.

### 2.4. Qualitative Methods

As noted, open-ended responses were specifically designed to explore topics that were not covered in the quantitative survey. The responses to these three questions ranged from 23 to 57 responses. To analyze participants' responses, the authors applied thematic template analysis methods [13]. Two of the authors (B.J., N.C.) read through all participants' responses and noted themes emerging from the data that consisted of participant's lived experiences. The qualitative data from the social workers' responses to the three open-ended survey questions were analyzed using template analysis. This method was chosen because the first-level codes were decided a priori by the topics in the questions. Other codes were added during the data analysis to incorporate emerging themes and patterns. Coding was completed by two of the study leads (B.J., N.C.) and was done by hand. Study authors reviewed the text and then met multiple times to identify second level codes, when to apply codes, and to develop a coding strategy. The coders utilized the codes to individually code all data and then met to discuss coded responses and to reach consensus on codes. Last, the coders chose salient themes and representative quotes. Second-level codes were developed using a constant comparative analysis of the data to elicit themes. Initial coding revealed 20 second level codes that were further analyzed and collapsed into 11 second level codes (see Table 1). Coding decisions were verified and checked by two of the authors (B.J. and N.C.) who compared and enhanced each other's codes comparing them to each other and the data until agreement was reached. Strategies to enhance rigor of the design included keeping an extensive audit trail, peer debriefing, and methodological and data triangulation with the quantitative results from the original study.

**Table 1.** First and Second Level Codes.

| First Level Codes | Second Level Codes |
| --- | --- |
| Experiences that Stay with you | Moral suffering<br>Fear for oneself and family<br>Seeing patient's and family's grace and resilience<br>Structural challenges/Lack of support |
| Wisdom Gained | Need to be calm, adaptable, and creative<br>Take care of self and family<br>Practice self-compassion and grace |
| Impact on Work | Isolation and powerlessness<br>Facing death and loss<br>No return to normal<br>Belief in one's own ability and resilience. |

### 3. Results

Of the 101 participants, fifty-seven provided open-ended responses to at least one question. The range of responses for each question was 23–57. Individual responses varied in length from short responses such as "Stay calm and always practice self-care" to extensive multi-sentence reflections on their professional challenges, personal losses, coping strategies, wisdom gained and ways to take care of themselves. Three key themes were identified a priori from the questions: experiences that stay with you; wisdom gained; and impact on work. First and second level codes are described in Table 1. Additional illustrative quotes are in Table 2.

**Table 2.** Additional Illustrative Quotes from Second Level Codes.

| Experiences That Will Stay with You | Illustrative Quotes |
| --- | --- |
| Moral suffering | "Not feeling good about any decisions. I have had to make over the past 8 months. No easy/right choice. Decision fatigue." |
| Fear for oneself and family | "Being afraid to be near my daughter who was working on a COVID 19 floor and her feeling like she could infect me. She and I were literally afraid to be near each other in the beginning." |
| Seeing patient's and family's grace and resilience | "I'm left without words or a true understanding of the mystery of the human spirit." |
| Structural challenges/Lack of support | "I will never forget the sudden realization that the management in our hospital system viewed social work as one of the first lines to go if the budget went south." |

| Wisdom Gained | Illustrative Quotes |
| --- | --- |
| Need to be calm, adaptable, and creative | "Stay calm! We are supposed to be the calm ones in a stressful situation, and this doesn't change with COVID." |
| Take care of self and family | "I don't think that promoting self-care is helpful in this situation, I think the shift should be to self-protection and help with boundary setting and balance. Never before has personal life been so intertwined with work life regarding safety, recognition that everyone has a unique circumstance and has unique needs during this time and need major mental health support for staff. Staff are incredibly adaptable and speaking up is helpful (always), but especially during an unprecedented time when creativity is needed, and all ideas may be more welcomed." |
| | "Take one day at a time. Advocate for yourself. Give yourself grace. Keep in contact with social work friends outside of the institution you work within to gain support outside the system as well." |
| Practice self-compassion and grace | "Social workers are always at the forefront of problem solving and critical thinking, whether during a pandemic or mass movements of fighting against racial injustice. Our patients/clients and our colleagues will always look to us for comfort, answers, hope and solace. It's something I am always grateful for; however, I know it also comes with the price of having to be very self-aware of my own limits and boundaries and to allow myself grace and compassion when I need to take time to step back from being a helper." |

| Impact on Work | Illustrative Quotes |
| --- | --- |
| Isolation and powerlessness | "I really worry about the effects of the isolation.""My biggest challenge has been isolation from family (elderly parents) and worry for everyone's safety." |
| Facing death and loss | "COVID-19 restrictions were such a mixed blessing. one week before we were sent home my stepfather died and on the day of his funeral his nursing home closed to visitors. My mother felt so overwhelmed with everything and since I was now working from home, I was able to bring her to my house for long afternoon breaks from her space. My time at home allowed me unexpectedly to provide emotional support to her during this sad time. I am not sure we would have been able to be together as much if my work schedule had not changed.""If appropriate, I may hug a mom of one of my patients, hold a hand, rub their back if they are upset. COVID does not allow for this personal touch anymore and I miss that." |
| No return to normal | "In our situation it has changed our lives almost completely and forced moving in new directions-wisdom: life continues-" |
| Belief in one's own ability and resilience | "This pandemic, amidst such a volatile political time, has made me even more fiercely unapologetic about fighting for the needs and rights of my patients and my peers. It's highlighted the inequity in our healthcare system and has reinforced my passion for advocating for the under-served and the forgotten." |

*3.1. Experiences That Will Stay with You, Always*

The question, "Can you tell us about an experience you have had related to COVID-19 that you feel will stay with you, always?" yielded 4 second-level codes: Moral suffering,

Fear for oneself and family, Seeing patient's and family's grace and resilience, and Structural challenges/Lack of support.

Participants described facing moral suffering as well as fear for their themselves and their families. Participants reported the most extreme moral distress in not being able to stop the suffering or provide the care they wanted to. For example, one participant described, "In my time covering other units, I had a teen whose entire nuclear family was infected. I spoke to the mom via telephone, unbeknownst to me, her mother had passed away that morning from COVID, and her father was in the ICU. Meanwhile, she had it, and her son was in our PICU. The raw shock and sadness were jarring. And not knowing what to say was hard, personally". Participants also described the moral suffering of not being able to allow families, including siblings, to be present at end of life as described here, "Yes—not having a sibling be able to come in and say goodbye until the patient was significantly physically decompensating at which point the parents did not feel comfortable exposing the sibling".

This moral suffering occurred in conjunction with a fear for their own health and safety and that of their families. Participants described the tension of wanting to provide care that they knew would increase their own personal risk such as, "Learning that I was exposed to a patient who tested positive! Providing bereavement support to people who were openly crying, not wearing masks, etc. and just having to hope and pray that they were not positive. (Even though I had on PPE)".

Participants were also profoundly impacted by the strength of the patients and families they cared for while simultaneously facing structural challenges and a lack of support for their role as social workers. Most reported a sudden and overwhelming change to their work experience and to their ability to define their own roles. They strived to find the opportunities to create meaning and care for patients who were struggling even before the pandemic. Patient and family distress was exacerbated during the pandemic by extreme isolation, barriers to care and lack of support. Despite these incredible stressors, pediatric oncology social workers described the power of seeing patients and coworkers' resilience and grace as illustrated by the quote, "Working with cancer patients/families includes very privileged moments; however, seeing the grace and resilience of families experiencing 2 compounding crises is at times more than I can I truly integrate. Participants also described being moved by their interprofessional colleagues. "There is no one experience. It is the cumulative experience of being with the peds onc nurses I have worked with for 15 years who have showed up every day for work and have continued to care for the kids just like it was any other day only with PPE". Observing the tenacity of their colleagues inspired the participants to continue their own work as this participant describes, "The resiliency of people (co-workers, patients, families)—we've adjusted and shifted a million times and will keep doing so".

The challenges of this ongoing grief and trauma were further exacerbated by structural challenges and lack of support from healthcare administration. A participant remarked, "Sadly, our physicians have not been understanding of social work (and care coordinators) working part-time from home. They want us there physically. They want our support which is great, but they have been less than nice about us working from home. It saddens me more than makes me feel valued".

### 3.2. Wisdom Gained

The question, "What wisdom have you gained from living and working through the pandemic that would be helpful to future social workers dealing with similar situations?" yielded three distinct second-level codes: Need to be calm, adaptable, and creative, Take care of self and family, and Practice self-compassion and grace.

Participants reflected that this crisis helped them gain a great deal of insight and wisdom about themselves, their priorities, and their own strengths as professionals. Many seemed to be able to adapt and identify ways to stay calm, creative, and even positive in the face of such stress. One participant shared, "On the positive side, processing and talking

with close family, friends, staff has allowed for good support and information sharing. I think that as social workers, we are uniquely trained to pivot on a regular basis, and so I really feel like my training served me well. It's been hard, and still is, but remembering that there is still a lot we do have control over, and much we can do to take care of ourselves and those we are close to". Again and again, participants described this ability to stay calm and be adaptable as a distinguishing characteristic of their profession that served them well during this crisis.

Facing so much trauma and carrying the emotional weight of pediatric oncology care put a strain on social workers' professional and personal lives. However, many reflected on the essential need to take care of themselves. One participant stated, "Rest and self-care is not selfish, and your team can keep you afloat. Humor is necessary. Togetherness (by whatever way possible) and keeping your greater purpose in mind is helpful. Feeling a part of something bigger than yourself is important".

Along with self-care, participants reported offering themselves grace and self-compassion during the traumatic times. One social worker offered, "Sometimes changes will occur unexpectedly, and it won't be within your power to fix the problems, so be kind to yourself and remember what it is that you can and cannot control and influence, and more importantly, remember that you're a human too, so you are not immune to the effects of difficult situations. You're allowed to feel your feelings, and it's okay if you aren't always on your A game".

### 3.3. Impact on Work

The third open-ended question, "Other ways that your work has been impacted by COVID-19 that have not been captured in this survey" yielded 4 second-level codes: Isolation and powerlessness, Facing death and loss, No return to normal, and Belief in one's own ability and resilience.

Even while facing multiple deaths, significant losses, and collective trauma, social workers were able to identify how to make meaning out of the work and recognize that there would be no return to "normal". The isolation was particularly distressing for some participants. "This has been very isolating. I live alone and working from home has been extremely lonely". Others described additional losses for their patients, "There is grief for things lost: in-person summer camp for our patients, our in-person, annual memorial service, patients not being able to have travel wishes. The gift of time is needed to process the losses that are obvious and those that may not be".

Social workers recognized that work will not simply return to how it was before COVID-19. One participant shared, "I mentioned that I don't think our work will return to "normal". I want to elaborate that I think COVID will reshape our future. I hope that we start implementing masks as a norm when we are ill. I also see virtual meetings becoming normalized and opportunities for virtual conferences to extend to people across the globe. I don't necessarily think not returning to normal is a bad thing".

Many participants noted their reinforced belief in the strengths and abilities of social workers as stated here, "Increased creativity to meet more patients, find available support to those that do not live locally, new ways of connection." This sense was often expressed as gratitude as shared by this participant, "Experience of gratitude for life, health and loved ones. It was always there but heightened now".

## 4. Discussion

Through free text open-ended questions, our study produced data that provided a rich picture into the experiences of pediatric oncology social workers during the first year of the COVID-19 pandemic. Throughout the responses, data that emerged highlighted both unprecedented personal and professional struggles as well as remarkable resilience.

Experiences that the participants believed will stay with them always centered around an often-immense feeling of personal isolation and moral suffering, including the seemingly unending witnessing of trauma and loss. Moral suffering occurs when providers are

constantly exposed to pain and suffering [14]. While social workers expect to be confronted with suffering as part of their job, it had previously rarely involved a threat to their own health. They also did not want to risk taking the virus home to their families and many described contemplating whether to continue in a job they loved or leave their positions in order to keep their families safe. This was especially relevant for social workers with young children or elderly family members.

The participants described having to sacrifice the natural response for human touch, including holding the hands of those who are frightened or crying. Having to provide care with a face shield, mask, and gloves (or from behind a computer screen) was especially difficult when working with critically ill or dying children. An additional challenging and unprecedented role for social workers, was having to enforce visitation limitations, and then witness children die without their whole family present. Limitations on sibling visiting presented a particular challenge for social workers who understand the need for sibling involvement at the end of life. This was seen as an enormous burden to bear and a type of moral suffering in which moral beliefs and situational safety conflict.

Social work as a profession is poised to encourage families and teams to identify strengths and help reduce anxieties [15] and the responses to the open-ended questions clearly spoke to the witnessing of considerable strength and tenaciousness of their patients and colleagues. Yet, just as social workers are instrumental in helping families build resilience in times of crisis and adversity, a sense of personal growth and wisdom emerged for them as well. The participants eloquently described needing to change their home life (telework), change how they practiced (virtual individual, family, and group sessions), change how they think about self-care, and consider changes in how they will lead their lives post pandemic. This speaks to an ability to identify and act upon their own resilience while navigating the demands of the COVID-19 pandemic [8]. Examples included exploring opportunities for broadening practice modalities, improvising and adapting, and making use of peer-based learning. Each of these has the potential to result in innovation and the capacity to transform clinical practice and workplace culture [9,15,16].

Similar themes of advocating for families, finding new meaning in one's work, and evaluating social work practices during times of crisis emerged during the coronavirus SARS-CoV-1 outbreak in 2003–2004 and in response to public tragedies [17–19]. How does COVID-19 compare to other pandemics or crises? It may be too early to really know. While it seemed that SARS-CoV-2 was moving toward endemicity, at the time of this writing, infections in the US are again rising. What we can expect is that pediatric oncology social workers may be at risk for experiencing "long-haul" impacts on how they practice and how they chose to balance their personal and professional lives. A recently published prospective cohort study found that anxiety associated with the pandemic was similar to the reported prevalence of anxiety following a diagnosis of pediatric malignancy [20]. This suggests that pediatric social workers will likely be addressing these long-term effects for some time to come [20]. It is critical that healthcare and educational institutions recognize the personal and professional suffering of COVID-19 on pediatric oncology social workers and develop strategies to support them.

This study has several important limitations. Only 56% of the participants chose to answer any of the open-ended questions. We cannot address whether those who chose to respond to the open-ended questions differed in terms of demographic factors from those who did not because we did not collect such data. Those who chose to answer these questions may not represent all pediatric social oncology social workers or generally shared perspectives. Next, the study was conducted during the first year of the COVID-19 pandemic and therefore, unique or continued challenges that presented later in the pandemic were not captured.

The pandemic disproportionately affected racial/ethnic minorities and other historical marginalized groups. This was not fully realized yet early in the pandemic and clearly is something to be explored and addressed in future research exploring the experiences of

all healthcare workers. Additionally, this was a US-based study and the experiences of pediatric oncology social workers in other parts of the world, were also not captured.

## 5. Conclusions

It is becoming increasingly understood how work-related and personal distress, including moral injury and burnout, are at unprecedented levels for healthcare providers [21–24]. The qualitative findings that emerged from this study highlights both the stressors and strengths of pediatric oncology social workers early in the pandemic. Capturing its impact through the pediatric social workers who were on the frontline caring for patients early in the pandemic, can serve to help other healthcare providers in the future.

Findings from this study are consistent with a recent study of healthcare social workers that similarly found themes of both moral suffering and professional resilience [25]. In this study, the authors found three main themes, "the hardest year of my career; the collective loss of our normal; and we were built for this". Social workers are uniquely trained in both trauma response and systems level change and so are uniquely qualified to provide leadership in responding to crisis at both the institutional and individual level. Taking what we have learned from the voices of social workers in this cohort and compiling our knowledge with others studied during this time will enable us to identify a model to help educate and sustain social workers long term.

There are cumulative lessons and wisdom to share from those who have worked through and survived COVID-19, like those social work professionals who worked during the early years of the AIDS crisis, or 9/11. Combining our knowledge and addressing global pandemics or crises as a part of social work training and education will help social workers, who are called into action during these times, to be there to meet the needs of the populations they serve. Recognizing the suffering of others while tending to their own cumulative grief creates a parallel paradigm for social workers and encourages a robust dialogue about who needs what from whom and how to support the social workers who are in the midst of the suffering.

**Author Contributions:** Conceptualization, L.W., B.J., N.C., A.F. and W.P.; methodology, B.J, L.W.; formal analysis, B.J. and N.C.; writing—original draft preparation, B.J., N.C., W.P. and L.W.; writing—review and editing, B.J., N.C., W.P., A.F. and L.W. All authors have read and agreed to the published version of the manuscript.

**Funding:** This research was supported, in part, by the Intramural Program of the National Cancer Institute, National Institutes of Health.

**Institutional Review Board Statement:** The Office of Human Subjects Research Protections at the National Institutes of Health determined that the survey format and content qualified as exempt from full Institutional Review Board review.

**Informed Consent Statement:** Participant consent was waived due to voluntary participation in survey.

**Data Availability Statement:** The data presented in this study are available on request from the corresponding author.

**Acknowledgments:** We would like to thank the pediatric oncology social workers for participation in this study and APOSW for allowing this research to be posted on their listserv. We would also like to acknowledge Marisol Corbitt who helped with the creation and formatting of the tables.

**Conflicts of Interest:** The authors declare no conflict of interest.

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
