# Peer review of "Reflections of Moral Suffering, Resilience, and Wisdom of Pediatric Oncology Social Workers during the COVID-19 Pandemic"

_curroncol, doi:10.3390/curroncol29090485_

Round 1
Reviewer 1 Report
Dear Authors this manuscript is well write. In the introduction, the background is sufficient and include all reference. The research design is appropriated to aims of paper. The methods adequately described and the results clearly presented.
The English language and style are fine.
Author Response
In this paper, the authors seek to explore the impact of the COVID19 pandemic on the experiences of pediatric oncology social workers. This is an important topic that has important implications for oncology HCP and therefore it is of interest to the special issue in Current Oncology.
Thank you.
The study design included an online survey distributed to participants recruited through the APOSW. The survey included quantitative and open-ended questions that were analyzed using thematic analysis.
I have two major concerns:
- The response rate for the open-ended questions was low. 23-57 respondents of the 101 who participated in the study completed the open-ended questions. The low response rate poses threat to the generalization of the study findings. I recommend that authors include an analysis of the demographic characteristics of participants who completed the open-ended questions and those who did not to see if there are key differences that may impact the interpretation of the results. I would also recommend addressing the low response rate to in the limitations section.
We thank the reviewer for this important comment. We agree that the response rate to the open-ended questions may not represent all the participants in this study. Unfortunately, we cannot address the sample’s specific demographics or diversity question because we did not collect this data. As noted, as we did not collect identifiers, this study was determined to be exempt from full IRB review. We have now clearly identified this important point as a limitation of the study. Additionally, while the average response rate for online surveys is closer to 44%, it is also possible that the lower response on the qualitative questions may reflect the distress and burden reported by these participants.
- The pandemic disproportionately affected racial minorities and other historical marginalized groups. I find the lack of consideration of these possible inequities as they shape the personal and professional experiences of social workers perplexing and invite the authors to comment on this should it come up in the data they collected, or at a minimum in the Limitations section.
This is another excellent point. We did not specifically ask about the inequities that these social workers experienced or witnessed. However, their responses indicated that they did experience Inequities as members of the healthcare team and that they witnessed suffering that was at times inequitable. This study was also administered early in the pandemic, and I am not sure we realized how disproportionately affected racial minorities would be yet. We have now added this important point to the limitations section as well.
Minor comments
- The paper only reports on the qualitative analysis. I therefore, would recommend clarifying this throughout and removing the details on the quantitative information that was collected (authors can indicate it was part of a larger study- and briefly describe other parts of survey.
Thank you. We have clarified in the Methods and Conclusion sections that this was part of a larger study. We have now removed other questions that were included in the larger study – specifically about questions pertaining to COVID-19 Exposure.
- Add a table with demographic information of the participants who completed the open-ended questions.
As noted above, as we did not collect demographic information about the participants (to avoid identifiers), we cannot create this table. This is now listed as a limitation.
- It is not clear what was the theoretical/empirical/clinical rational for the three open questions that were presented to participants. It would be helpful if authors can provide more context/clear rationale for this choice.
Thank you for this important question. Mixed methods research allows us to have both quantitative data for numbers and qualitative data for context. We felt that the data that emerged from these open-ended questions might add deeper insights and a more complex picture of the lived experiences of these providers. Specifically, asking about an experience during COVID-19 that has had a significant impact and “will stay with you always” would help us learn about experiences that were unique to the COVID-19 pandemic. Asking for advice (“wisdom gained”) could be helpful to other pediatric oncology social workers as they re-entered in-person care. Furthermore, leaving an open question, about other ways one’s work has been impacted by COVID-19, could allow us to learn about anything important, that we might have missed in the larger study. We have now cited how these questions had previously been used with pediatric palliative care providers with great success.
Reviewer 2 Report
In this paper, the authors seek to explore the impact of the COVID19 pandemic on the experiences of pediatric oncology social workers. This is an important topic that has important implications for oncology HCP and therefore it is of interest to the special issue in Current Oncology.
The study design included an online survey distributed to participants recruited through the APOSW. The survey included quantitative and open-ended questions that were analyzed using thematic analysis.
I have two major concerns:
1. The response rate for the open-ended questions was low. 23-57 respondents of the 101 who participated in the study completed the open-ended questions. The low response rate poses threat to the generalization of the study findings. I recommend that authors include an analysis of the demographic characteristics of participants who completed the open-ended questions and those who did not to see if there are key differences that may impact the interpretation of the results. I would also recommend addressing the low response rate to in the limitations section.
2. The pandemic disproportionately affected racial minorities and other historical marginalized groups. I find the lack of consideration of these possible inequities as they shape the personal and professional experiences of social workers perplexing and invite the authors to comment on this should it come up in the data they collected, or at a minimum in the Limitations section.
Minor comments
1. The paper only reports on the qualitative analysis. I therefore, would recommend clarifying this throughout and removing the details on the quantitative information that was collected (authors can indicate it was part of a larger study- and briefly describe other parts of survey.
2. Add a table with demographic information of the participants who completed the open-ended questions.
3. It is not clear what was the theoretical/empirical/clinical rational for the three open questions that were presented to participants. It would be helpful if authors can provide more context/clear rationale for this choice.
Author Response
Thank you!
Round 2
Reviewer 2 Report
Thank you for sending me the revision for review. I believe the authors addressed the concerns in a satisfactory manner